# Supramammillary glutamate neurons are a key node of the arousal system

Nigel P. Pedersen [1], Loris Ferrari[2,3], Anne Venner [2,3], Joshua L. Wang [2], Stephen B.G. Abbott[2], Nina Vujovic[3], Elda Arrigoni[2,3], Clifford B. Saper[2,3] & Patrick M. Fuller[2,3]

Basic and clinical observations suggest that the caudal hypothalamus comprises a key node of the ascending arousal system, but the cell types underlying this are not fully understood. Here we report that glutamate-releasing neurons of the supramammillary region (SuM$^{vglut2}$) produce sustained behavioral and EEG arousal when chemogenetically activated. This effect is nearly abolished following selective genetic disruption of glutamate release from SuM$^{vglut2}$ neurons. Inhibition of SuM$^{vglut2}$ neurons decreases and fragments wake, also suppressing theta and gamma frequency EEG activity. SuM$^{vglut2}$ neurons include a subpopulation containing both glutamate and GABA (SuM$^{vgat/vglut2}$) and another also expressing nitric oxide synthase (SuM$^{Nos1/Vglut2}$). Activation of SuM$^{vgat/vglut2}$ neurons produces minimal wake and optogenetic stimulation of SuM$^{vgat/vglut2}$ terminals elicits monosynaptic release of both glutamate and GABA onto dentate granule cells. Activation of SuM$^{Nos1/Vglut2}$ neurons potently drives wakefulness, whereas inhibition reduces REM sleep theta activity. These results identify SuM$^{vglut2}$ neurons as a key node of the wake–sleep regulatory system.

[1] Department of Neurology and Epilepsy Service, Emory University, Atlanta, GA 30322, USA. [2] Department of Neurology, Beth Israel Deaconess Medical Center, Bostan, MA 02215, USA. [3] Division of Sleep Medicine, Harvard Medical School, Bostan, MA 02215, USA. Clifford B. Saper and Patrick M. Fuller contributed equally to this work. Correspondence and requests for materials should be addressed to N.P.P. (email: npeders@emory.edu) or to P.M.F. (email: pfuller@bidmc.harvard.edu)

In the 1930s, von Economo[1] proposed that damage to diencephalic-mesencephalic junction was responsible for marked somnolence in patients with epidemic encephalitis lethargica, and Ranson[2] was able to reproduce this somnolence with lesions of the caudal hypothalamus in monkeys. Supporting this observation, Moruzzi and Magoun[3] found that stimulation of the caudal hypothalamic region could elicit wakefulness and electrocortical low-voltage fast activity at a threshold comparable to brainstem arousal sites, and at a lower threshold than for the thalamus. This induced arousal was seen even after destruction of the thalamus[4], while caudal hypothalamic lesions or inactivation resulted in marked and often prolonged somnolence[2, 3, 5–10]. These experiments and similar observations in humans[1, 11–16] supported the existence of a key node of the arousal system within the caudal hypothalamus.

While orexin neurons in the lateral hypothalamus and tuberomammillary histaminergic neurons both promote arousal, lesions of these caudal hypothalamic cell groups have not been found to cause the level of hypersomnolence produced by larger posterior hypothalamic lesions[17, 18]. Based upon its anatomical connectivity and cellular composition, we hypothesized that the supramammillary nucleus (SuM), a region lying just dorsal to the mammillary body in the caudal hypothalamus, might comprise a key site for wake-promotion. The SuM provides extensive innervation of the cerebral cortex[19], wake-promoting basal forebrain[20–25], and the hippocampus[19, 20, 26–28, 33]. Neurons in the SuM are known to play a role in driving hippocampal theta rhythm, displaying theta band field potentials in freely moving rats[29], and unit discharges that correlate with this theta modulation[30, 31]. Previous work has examined the relationship of SuM activity to hippocampal local field potentials and learning[27, 32–34], and suggested a role for SuM neurons in REM sleep[13, 35], but the SuM has not been studied at all in relation to wakefulness, nor with respect to the possibly distinct roles of SuM subpopulations in REM sleep and wakefulness. Whereas most neurons in the

SuM region are glutamatergic[36, 37], there is anatomical evidence that some of these neurons may contain other neuromodulators[23, 38–41] including a subpopulation of SuM neurons targeting the hippocampus that contain anatomical markers of both GABA and glutamate release[42, 43]. Here we used a combination of targeted genetic and viral vector technologies in behaving mice to determine whether SuM neurons contribute to electrocortical and behavioral arousal. We then employed in vitro optogenetics and application of a novel non-parametric mapping technique to delimit and characterize three distinct cell populations in the SuM region, each of which differentially contribute to cortical and hippocampal activation and behavioral wake. Our results show that SuM$^{Vglut2}$ neurons are necessary for normal wakefulness, with activation potently promoting theta and gamma EEG activity and wakefulness. SuM$^{Nos1/Vglut2}$ neurons can promote wake when activated and are also necessary for normal REM theta activity. SuM$^{Vgat/Vglut2}$ neurons provide dense innervation of the dentate gyrus and CA2, monosynaptically releasing both GABA and glutamate on to the proximal dendrites of dentate granule cells, but only minimally affecting sleep−wake and,

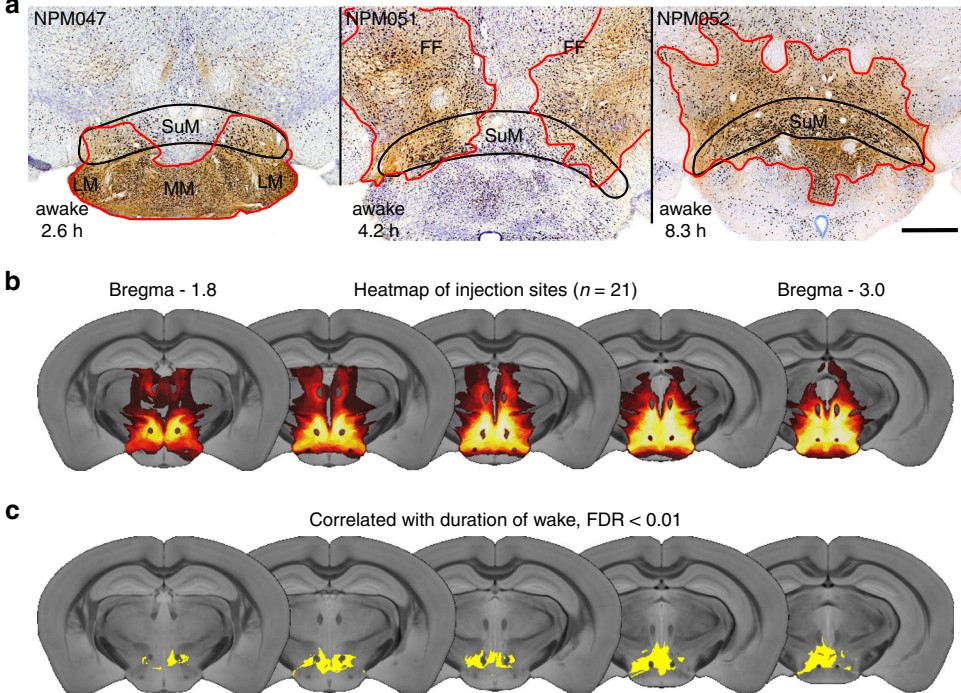

**Fig. 2** Wake-promoting neurons are within the SuM region. **a** Smaller injection sites (red outline) showing prolonged wakefulness (hours of wake after CNO at bottom left of each panel) was highest with involvement of the whole SuM (scale bar 500 μm). **b** Plot of all injection sites as a 'heat map' of injection overlap (n = 21, dark red minimal overlap (1 case), white maximal (18 cases)). **c** Non-parametric permutation analysis shows that the area of transduction correlated with the longest duration of wakefulness after CNO is the caudal hypothalamus, most notably the SuM (yellow area false discovery rate < 0.01). MM–medial mammillary, LM–lateral mammillary, and FF–fields of Forel

**Fig. 1** Caudal hypothalamic activation potently drives wakefulness. **a** Cartoon showing AAV-FLEX-hM3Dq-mCherry virus recombining in vglut2 neurons in the presence of Cre. **b** Example injection site showing diaminobenzadine (DAB) labeling of mCherry (red−brown staining, red outline) and nickel/cobalt enhanced DAB (black) labeling of cFos in nuclei (see also inset, scale bar 10 μm) of neurons activated by CNO (main panel scale bar 500 μm, SuM shown in black outline). **c, d** Hypnograms after vehicle and CNO showing prolonged wakefulness after activation of glutamate neurons (W–wake, N–NREM sleep, R–REM sleep; gray = dark period). **e, f** Compressed spectral array (CSA) based on fast Fourier transform (FFT) after vehicle and CNO (0–100 Hz, black bar shows the dark period, short white bars represent 20 min and are spaced 1 h apart; time periods in blue and red boxes for NPM005 in **c** and **d** are shown in **e** and **f**), with prominent theta activity after CNO. **g** Group data (mean ± SEM) showing significantly higher wakefulness for nine hours after CNO (2-way repeated measures ANOVA for treatment $p < 0.0001$, Bonferroni ***$p < 0.001$; *$p < 0.05$). **h** Waking EEG (mean ± SEM) shows increased high theta (HTheta, 7−13 Hz, $p = 0.0148$), gamma (30−120 Hz, $p = 0.0005$), and decreased low theta component of the FFT power spectrum (LTheta, 4−7 Hz, $p = 0.0027$). **i** Cortical and hippocampal (inset) EEG FFT spectra (log frequency, arbitrary units (AU)) showing prominent hippocampal ~9 Hz activity. **j** Raw wake EEG and EMG data with CSA (0−30 Hz) over 5 s for vehicle, and **k**. CNO, with prominent 9−10 Hz activity visible in the CSA in the latter case

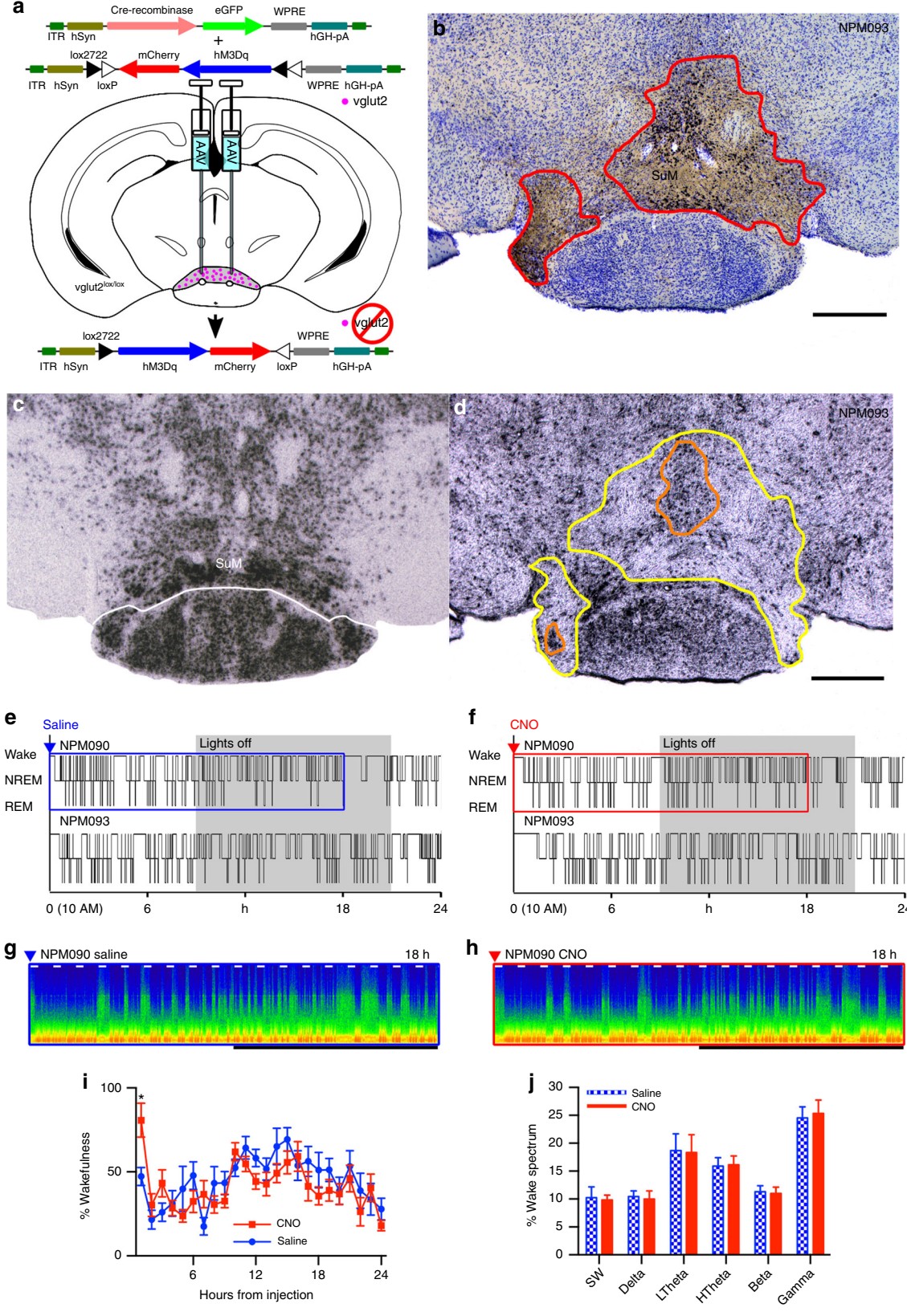

interestingly, do not as potently promote theta EEG activity despite targeting the hippocampal network. In concert with these observations and the described anatomical connectivity of the SuM, these results suggest that SuM$^{Vglut2}$ neurons comprise a key nod of the sleep–wake regulatory system that is positioned to control both the hippocampal system and arousal.

## Results

**SuM[vglut2] neurons drive wakefulness.** In our first experiment, we tested whether glutamate neurons of the caudal hypothalamus drive wakefulness. In order to selectively activate caudal hypothalamic glutamate neurons, including SuM glutamate (SuM[vglut2]) neurons, we first made large stereotaxic injections of an adeno-associated viral vector (AAV) that conditionally expressed an excitatory modified human G-protein coupled muscarinic receptor (hM3Dq)[44, 45] fused to the mCherry fluorescent protein (AAV-hM3Dq), into the caudal hypothalamus of *vglut2-IRES-Cre* mice (Fig. 1a). Functional transduction of the SuM neurons was shown by administration of the hM3Dq ligand clozapine-n-oxide (CNO, 0.3 mg/kg in 10 µl/g normal saline vehicle i.p.), which drove cFos expression in the transduced neurons (Fig. 1b, upper right). Administration of CNO after injection of AAV-hM3Dq in non-Cre-expressing control littermates, or injections of vehicle (Fig. 1c) after injection of AAV-hM3Dq in Cre-expressing mice was without effect on sleep–wake or cFos expression. In contrast, CNO-driven activation of hM3Dq-expressing SuM[vglut2] neurons potently promoted continual wakefulness, in some cases for periods as long as 12 h, during the animals' normal sleep period (Fig. 1d; administration 3 h after lights-on (Zeitgeber (ZT) 3, 10 AM; Latency to sleep: saline $22 \pm 4$ (SEM), CNO $534 \pm 54$ min, paired *t*-test, $p = 0.000014$, $n = 9$, see group data Fig. 1g). In these mice, CNO also caused a significant increase in higher range theta activity (HTheta 7–13 Hz, paired *t*-test with Bonferroni correction, $p = 0.0148$, $n = 5$, spectral peak at about 9 Hz), and decrease in lower frequency theta EEG activity (low range theta (LTheta) 4–7 Hz, $p = 0.0027$) (Fig. 1e, f, h, i) with a significant increase in gamma frequency EEG activity (gamma 30–120 Hz, $p = 0.0005$). Bipolar recording from the hippocampus ($n = 6$) revealed similar EEG spectral changes, confirming concomitant hippocampal activity with a spectral peak at 8.5–9 Hz (Fig. 1i upper right). Importantly, the repertoire of behavior after CNO in these mice was normal (Supplementary Movie 1), and the amount or speed of locomotor activity did not account for theta activity (mice were often stationary, e.g., see raw data in Fig. 1j, k with periods of low EMG in both examples, but increased theta power (red) in Fig. 1k; Supplementary Movie 1). Given the known role of the SuM in the genesis of the theta activity, we hypothesized that the wake-promoting cell group was either identical with or adjacent to the SuM neurons that promote theta.

We next scattered small injections of AAV-hM3Dq throughout the SuM and adjacent regions to better understand the precise location of the wake-promoting neurons ($n = 15$). We studied each injection site in detail, noting which nuclei were wholly or partially involved in the injection site and their degree of transduction. We noted that small injection sites flanking the SuM were associated with minimal wakefulness after CNO injection, while those involving the whole of the SuM were associated with the most prolonged wakefulness (see examples, Fig. 2a). To analyze these data in an objective fashion, we made a three dimensional (3D) map of the injection site, plotted as 30 µm cubic voxels on to a template derived from a high resolution mouse brain MRI scan (Fig. 2b). We then used a non-parametric permutation test of voxels containing transduced neurons and correlated this with the duration of wakefulness resulting from SuM activation (Fig. 2c, $n = 21$; Methods section). This produced an anatomical *z*-score, revealing that the combination of voxels that was statistically associated with increased wakefulness corresponds best to the SuM region (false discovery rate < 0.01 and corresponding to a $z \geq 3.216$, $n = 21$). Conversely, caudal hypothalamic regions immediately surrounding the SuM region did not significantly contribute to prolonged wakefulness. Involvement of somewhat further cell groups such as the lateral and perifornical regions may have accounted for the most prolonged duration of wakefulness with the largest injection sites, given the known wake-promoting effect of orexin neurons that also express vglut2 in that region. Smaller injections with transduction constrained within the SuM resulted in just over 5 h of wakefulness (saline $29 \pm 5$ min versus CNO $310 \pm 40$ min, $n = 6$, $p = 0.0012$), indicating that this caudal hypothalamic cell group independently promotes wakefulness.

**SuM actions depend on glutamate.** Given the presence of other mediators such as nitric oxide, GABA, and several peptides[23, 38–43] in the SuM, we next sought to determine whether glutamate is necessary for the wake-promoting effect of SuM neuronal activation. To test this hypothesis, and whether the action of unopposed GABA release might decrease wakefulness, we placed stereotaxic injections of both AAV-Cre and a conditional AAV-hM3Dq into the SuM region of *vglut2[lox/lox]* mice, to concurrently disrupt synaptic glutamate release and express hM3Dq in transduced neurons (Fig. 3a). Because only neurons that had been transduced with the AAV-Cre (and therefore deleted the *vglut2* gene) would also express the hM3Dq, CNO would only drive neurons that lacked functional VGLUT2 protein. Histologically, there was effective co-expression of hM3Dq and cFos after CNO over a large part of the SuM (Fig. 3b), with correlated loss of *vglut2* in situ hybridization signal (compare Fig. 3c, d). Activation of these SuM neurons resulted in a brief period of wakefulness that was not statistically significant from vehicle injection (vehicle $21 \pm 4$ (SE) min versus CNO $49 \pm 12$ min, $p = 0.06$, $n = 6$, Fig. 3e–h, summary data Fig. 3i), and was far shorter than comparable activations of neurons without vglut2 deletion (Fig. 1c, d), suggesting that glutamate release from SuM neurons is required for the bulk of the wake-promoting effects of SuM activation. Furthermore, EEG spectral changes associated with activation of SuM vglut2-only neurons were abolished, with no changes in individual spectral bands in any state ($p > 0.99$ in wake, $n = 6$, see wake spectra Fig. 3g, h, j).

**Fig. 3** Wake-promotion after SuM activation depends on glutamate. **a** Cartoon showing that after co-injection of AAV-FLEX-hM3Dq-mCherry and AAV-Cre in a vglut2flox mouse, that vglut2 is disrupted in neurons expressing Cre, with Cre positive neurons also able to drive hM3Dq expression. Thus, the only neurons activated by CNO are neurons that lack glutamate release. **b** mCherry-like immunoreactivity and cFos expression after CNO administration (left) showing where hM3Dq was transduced, and cFos expression consistent with cellular activation after CNO. **c** Normal expression of vglut2, revealed by in situ hybridization, and **d** in situ hybridization showing the loss of vglut2-expression in the injection site in the same mouse shown in **b** (area with near-complete signal loss shown in yellow, area with deletion in some neurons is outlined in orange). **e**, **f** Hypnograms of two mice showing marked attenuation of wake-promoting effects of SuM activation, consistent with substantial dependence on glutamate release. **g**, **h** CSA showing the effect of vehicle and the near-absent effect of activation with CNO in the period shown by the blue and red boxes in **c** and **d**, respectively (0–100 Hz, black bar shows the dark period, short white bars represent 20 min and are spaced 1 h apart). **i** Pooled data (mean ± SEM in 1 h bins, $n = 6$) after saline and CNO administration, with post hoc comparisons showing that more of only the first hour is spent awake after CNO (Bonferroni corrected comparison for 1 h $p = 0.0153$, all other hours $p > 0.99$). **j** Spectral changes associated with SuM activation are also abolished (wake from 6 h after injection shown, mean ± SEM, $n = 6$)

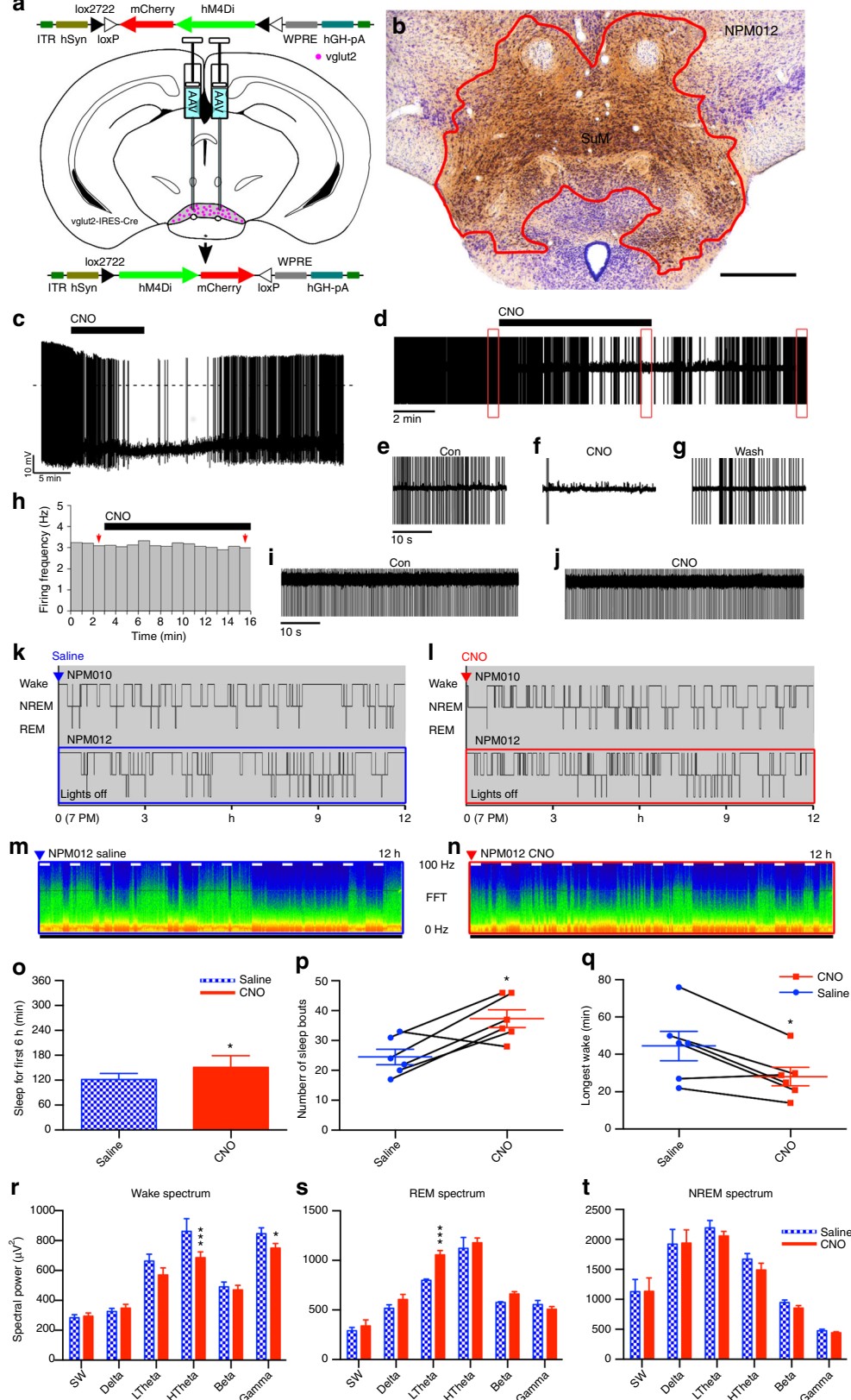

Chronic vglut2 deletion in the SuM had little effect on amounts of wake, REM, and NREM sleep, all of which remained within normal limits for mice (wake $45.2 \pm 2.5\%$, NREM $48.4 \pm 2.4\%$, REM $6.4 \pm 0.2\%$, $n = 6$). However, like the ibotenic acid lesions of the rat SuM by Renouard et al.[35], the deletions in these experiments never covered the entirety of the SuM, and because of the time it took for the full deletion to occur and for the animals to recover, there was ample time for compensation. We therefore tested the effect of acute inhibition of the SuM$^{Vglut2}$ cells on wakefulness.

**SuM$^{vglut2}$ neuronal inhibition causes somnolence.** To examine the effect of acute inhibition the SuM$^{Vglut2}$ neurons, we used a conditionally expressed modified inhibitory human muscarinic 4 receptor (hM4Di)[44, 45]. We made large injections of an AAV-FLEX-hM4Di-mCherry (AAV-hM4Di) vector in the posterior hypothalamus of *vglut2-IRES-Cre* mice (Fig. 4a), deliberately including as much as possible of the SuM$^{vglut2}$ cell population (Fig. 4b). We validated that CNO was able to inhibit transduced neurons by recording SuM neurons in vitro in the cell attached mode. CNO (5 µM) reduced the firing frequency of SuM$^{vglut2}$ neurons that express hM4Di (firing frequency in control 0.89 ± 0.27 Hz, CNO 0.09 ± 0.05 Hz; $p = 0.02$ paired $t$-test, $n = 6$) (Fig. 4c–g). CNO had no effect in SuM$^{vglut2}$ neurons that only express mCherry (firing frequency in control 1.2 ± 0.6 Hz, CNO 1.35 ± 0.65 Hz; $p = 0.42$, paired $t$-test, $n = 5$) (Fig. 4h–j).

In freely behaving mice, increased sleep and fragmentation of wakefulness and sleep were present after CNO injection (Fig. 4k–n). Increased NREM sleep was significant, elevated by about 20% over the 6 h following CNO administration (injection at ZT12, typical time of maximal wakefulness, saline 33.8 ± 2.80% (SEM) versus CNO 41.95 ± 1.49%, 2-way ANOVA, $p = 0.0199$, $n = 6$, Fig. 4o). This was at the expense of both wakefulness and REM sleep, although neither reduction reached statistical significance (2-way ANOVA by state; wake—62.4 ± 2.85 versus 55.8 ± 1.94%, $p = 0.0659$, $n = 6$; REM −3.31 ± 0.34 versus 2.31 ± 0.72%, $p = 0.2614$). There was an increased number of sleep bouts in the six hours following injection, as well as a reduction in the maximal time for which wakefulness was maintained (saline 24.5 ± 2.56 (SEM) versus 37.3 ± 2.90 bouts, paired $t$-test, $p = 0.0194$, $n = 6$, Fig. 4p), with the longest wake bouts after saline being significantly longer than after CNO (44.5 ± 7.83 versus 28.2 ± 4.98 min, $p = 0.0152$, $n = 6$, Fig. 4q). The spectral power changes were inverse to those of activation, i.e., we saw a significant decrease in high frequency theta and gamma power during wakefulness (HTheta $p < 0.0001$, and Gamma $p = 0.0472$, $n = 5$, Fig. 4r). There was also an increase in the power of low-frequency (4–7 Hz), but not high-frequency (7–13 Hz) theta during REM sleep ($p < 0.0001$, $n = 5$, Fig. 4s), with no effect on other REM spectral bands ($p > $ than 0.16 for all bands), nor in the power spectrum of NREM sleep ($p = 0.38$ to $> 0.99$ for all bands, Fig. 4t). These changes in power spectrum and sleep architecture are consistent with sleepiness during the wake state and the inability to sustain longer bouts of wakefulness, but normal NREM sleep.

**SuM$^{vgat/vglut2}$ neurons.** GABAergic neurons surround the mammillothalamic tract as it penetrates the SuM[37, 43], and spread medially and laterally in the more posterior SuM. The terminals of these neurons in the hippocampus have been reported to contain markers of both GABA and glutamate release[42, 43]. Given that these neurons are within our region of interest, we sought to

understand if they were also capable of driving cortical and hippocampal activation and/or behavioral arousal. We first sought to verify the exact location of the GABA/glutamate neurons in the SuM. We performed in situ hybridization for *vglut2* mRNA in a *vgat-IRES-Cre* reporter mouse (L10 reporter line, see Methods), revealing *Vglut2* expression in all of the GABAergic neurons surrounding the mammillothalamic tract in the SuM—thus neurons in the SuM are of either SuM$^{vglut2}$ or SuM$^{vgat/vglut2}$ phenotype (Fig. 5a). The latter neurons are larger than the adjacent ones expressing only vglut2 (20.54 ± 0.64 (SEM) µm longest axis of soma, versus 13.93 ± 0.50 µm, unpaired $t$-test, $p < 0.0001$, $n = 12,12$), and correspond with what has been termed the grandicellular field (SuMg)[40], forming a clear subset of larger neurons in the SuM (Fig. 5b). There were no examples noted of neurons that were only vgat positive within the SuM.

To determine whether these neurons co-release both neurotransmitters, we conditionally transduced channelrhodopsin-2 (ChR2) in the SuM of *vgat-IRES-Cre* mice (SuM$^{vgat}$; Fig. 5c), then made whole-cell recordings from dentate granule cells in hippocampal brain slices (Fig. 5d). Photostimulation of *Vgat* positive axons and terminals originating from the SuM (as shown in the cartoon in Fig. 3d) evoked fast postsynaptic currents in granule cells (15/18 neurons). Application of bicuculline or DNQX reduced the amplitude of, but did not eliminate, photo-evoked postsynaptic currents. The combination of bicuculline and DNQX completely blocked postsynaptic responses, suggesting mediation by release of both GABA and glutamate, respectively activating GABA-A and AMPA receptors (Fig. 5e). Selective blockade with these two antagonists revealed that GABA-A-mediated postsynaptic currents had typical slower kinetics than AMPA-mediated currents[46] (Fig. 5f; decay time constants: 34.57 ± 5.99 ms for the GABA-A-mediated current and 5.15 ± 1.06 ms for the AMPA-mediated currents, fitting with a single exponential; $n = 3$). Reversal potentials were −49.74 ± 3.65 mV (SEM) and +11.94 ± 2.02 mV ($n = 4$ neurons from four brain slices from four mice; $p < 0.001$, paired $t$-test), compatible with GABA-A-mediated chloride and AMPA-mediated sodium currents, respectively (Fig. 5g, h). Importantly, both the AMPA-mediated and the GABA-A-mediated photo-evoked postsynaptic currents had a short onset delay characteristic of photostimulation-evoked release (GABA-A-mediated: 5.04 ± 0.37 ms and AMPA-mediated: 4.33 ± 0.23 ms, $n = 8$, 9 neurons (from 3 vglut2-Cre and 5 vgat-Cre mice, respectively), $p = 0.002$, paired $t$-test) supporting monosynaptic release of both GABA and glutamate from SuM$^{vgat/Vglut2}$ terminals onto dentate granule cells (Fig. 5i, j). Overall, photostimulation of the SuM$^{vgat/Vglut2}$ → DG pathway evoked GABA-A-mediated and AMPA-mediated synaptic responses in a large proportion of dentate granule cells (82%) whereas it evoked solely GABA-A-mediated or AMPA-mediated responses in only 9 and 9% of granule cells ($n = 18$), respectively. We also conditionally expressed ChR2 in SuM$^{vglut2}$ neurons, with

**Fig. 4** Inhibition of the SuM results in somnolence and wake fragmentation. **a** Cartoon showing AAV-FLEX-hM4Di-mCherry recombining in vglut2 neurons that express Cre. **b** Conditional transduction of inhibitory hM4Di in vglut2-IRES-Cre mice (red−brown staining outlined in red; scale bar 500 µm). **c**−**g** CNO (5 µM) inhibits the firing of SuM$^{vglut2}$ neurons that express hM4Di-mCherry (**c** current-clamp recordings; **d** cell attached recordings; **e**−**g** expanded traces of the outlined region from **d**, show firing in control, during CNO application and in washout). **h**−**j** CNO (5 µM) has no effect on the firing of SuM$^{vglut2}$ neurons that only express mCherry (**h**; firing frequency, 1-min bin, red arrows indicate the 1-min bins represented in **i**, **j**; **i**, **j** cell attached recordings of the neuron in **h**). **k**, **l** Hypnograms from two mice over 12 h after vehicle and CNO, showing increased sleep and fragmentation of sleep and wakefulness after inhibition of the SuM. **m**, **n** CSA after vehicle and CNO from the periods shown by the blue and red boxes, respectively, in **c** and **d** (0−100 Hz, black bar shows the dark period, short white bars represent 20 min and are spaced 1 h apart). **o** There is increased overall NREM sleep in the 6 h after CNO administration (~20%; 2-way ANOVA for 6-hourly bins between vehicle and CNO, graph with mean ± SEM, $p = 0.0199$). **p** wake-sleep fragmentation with an increase in sleep bouts over the 6 h after CNO when compared with vehicle (paired $t$-test, mean ± SEM, $p = 0.0194$). **q** The length of the longest episode of wakefulness in the same time period was also significantly reduced (paired $t$-test, mean ± SEM, $p = 0.0152$). **r** Power spectrum (band mean ± SEM) with a significant suppression of waking high theta activity (2-way repeated measures ANOVA, Bonferroni ***$p < 0.0001$) and gamma activity (*$p = 0.0472$), and **s** an unexpected increase in low theta power during REM sleep (***$p < 0.0001$), with **t** no change in the NREM spectrum

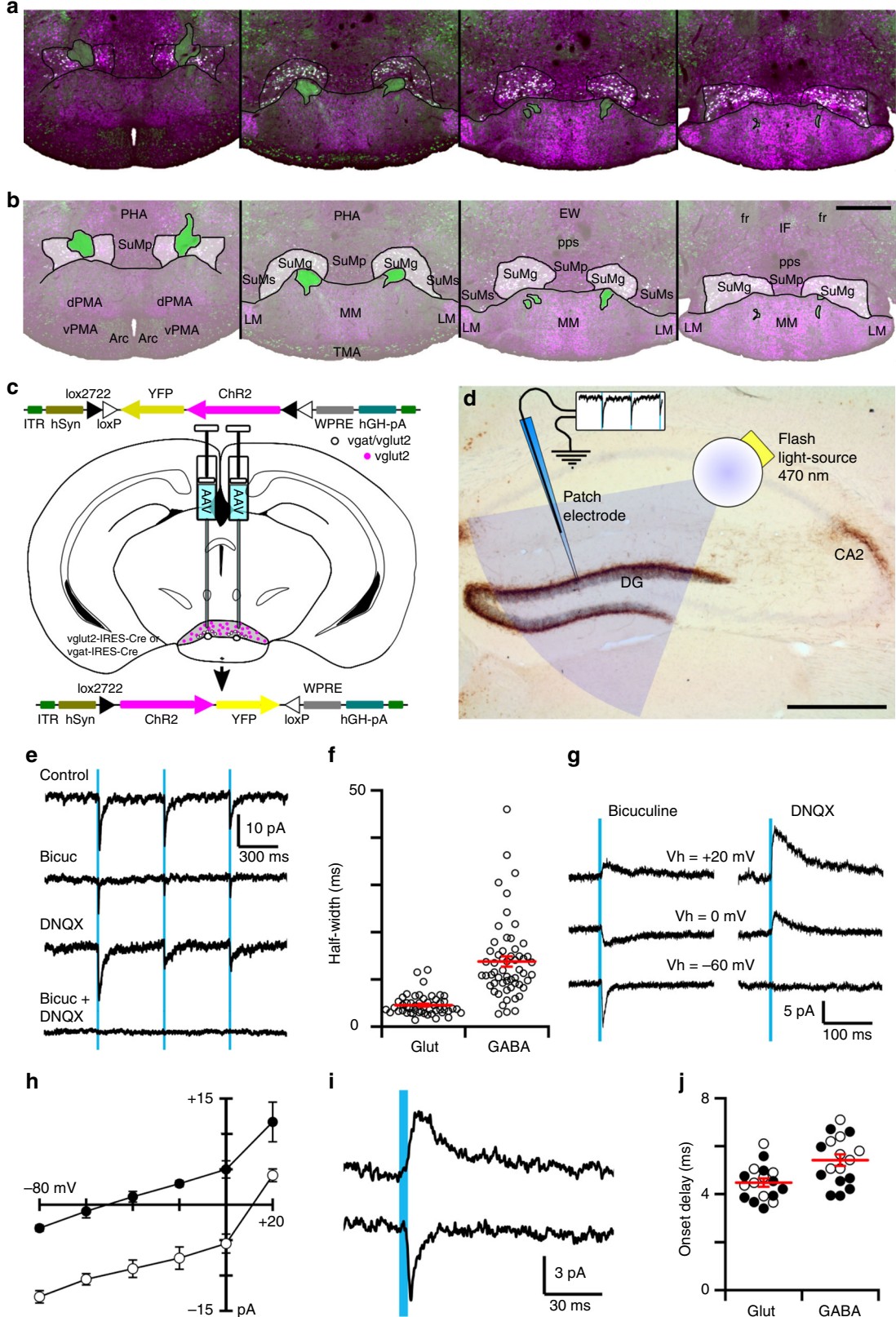

photostimulation of terminals again evoking both GABA-A-mediated and AMPA-mediated synaptic responses in 90% of dentate granule cells, with AMPA responses alone in 10% of granule cells ($n = 9$). This is consistent with previous studies indicating that the SuM hippocampal projection arises from the neurons adjacent to the mammillothalamic tract[19]. The mean amplitude and mean onset delay of the photo-evoked glutamate-mediated currents were $-5.36 \pm 1.06$ pA and $4.55 \pm 0.25$ ms (Vh $= -70$ mV) and for the GABA-A-mediated currents were $+7.11 \pm 0.89$ pA and $5.84 \pm 0.25$ ms (Vh $= 0$ mV; $n = 8$), again consistent

with a photic stimulation-evoked monosynaptic response (Fig. 5j). Together, these results suggest that the SuM input to the dentate gyrus predominantly originates from neurons that express both vesicular transporters for glutamate and GABA (SuM$^{vgat/vglut2}$) and that these SuM$^{vgat/vglut2}$ neurons co-release both transmitters at their synaptic terminals.

We next examined whether SuM$^{vgat/vglut2}$ neurons, representing about 15% of SuM neurons (see below), were wake-promoting, by activating these neurons in vgat-IRES-Cre mice after transduction with AAV-hM3Dq (Fig. 6a). Injection sites covering the region of vgat expression in the SuM had a modest but significant effect on the duration of wakefulness following CNO injection (saline $14 \pm 3.6$ (SE) min, versus CNO $92 \pm 10$ min, paired $t$-test, $n = 4$, $p = 0.0027$; Fig. 6c–g). Thus, activation of these neurons was significantly less wake-promoting than activation of the entire population of SuM$^{vglut2}$ neurons (duration of wakefulness, CNO in vgat-Cre mice $92 \pm 10$ min, vglut2-Cre mice $310 \pm 40$ min, $n = 4, 6$, $p = 0.0028$).

**SuM$^{Nos1/Vglut2}$ neurons drive wake.** As previously described in cats and rats[40, 41], the mouse SuM contains a subpopulation of Nos1-containing neurons (Fig. 7a, b; see also Allen Mouse Brain Atlas[37]). Given that cortical Nos1-expressing neurons have been previously implicated in behavioral state control, albeit in the regulation of slow-wave-sleep[47], we sought to further understand the anatomic distribution, cellular co-localization and physiology of SuM$^{Nos1}$ neurons. We first examined Nos1 immunoreactivity in Vglut2-Cre-L10 reporter mice, and found Nos1 immunoreactive neurons to be scattered throughout the SuM, representing about 15–20% of vglut2-positive neurons at all rostrocaudal levels (Fig. 7a, b). We then used Nos1-IRES-Cre mice and stereotaxic microinjection of AAV-hM3Dq into the SuM (Fig. 7c) to selectively activate these neurons with CNO (see cFos in transduced neurons, Fig. 7d). Activation of these neurons potently drove sustained wakefulness (latency to sleep after saline $33 \pm 4$ (SEM) min versus CNO $215 \pm 27$ min, $n = 7$, $p = 0.0007$, Fig. 7e, f), with post hoc comparisons (Bonferroni corrected) showing a significantly elevated wakefulness for four hours after CNO injection (Fig. 7i). While activation of SuM$^{Nos1/Vglut2}$ neurons resulted in lesser wakefulness than activation of all SuM$^{vglut2}$ neurons in the cases with large injection sites (Fig. 1), the duration of wakefulness was more than twice as long as after activating a similarly numerous population of SuM$^{vgat/Vglut2}$ neurons, and similar to the amount of wakefulness driven by activation of SuM$^{vglut2}$ neurons with smaller injection sites constrained within the SuM (see above).

**SuM$^{Nos1/Vglut2}$ neurons contribute to REM theta.** To examine the necessity of SuM$^{Nos1/Vglut2}$ neurons for normal wakefulness, we made stereotaxic injections of AAV-hM4Di into the SuM of Nos1-IRES-Cre mice (Fig. 8a), with transduction in the SuM$^{Nos1/Vglut2}$ cellular field (Fig. 8b). There was no significant difference in wake, NREM and REM sleep duration when mice were injected with saline (vehicle) or CNO (0.3 or 0.9 mg/kg) (Fig. 8c–h), no overall increase in sleep over the first 6 h ($p = 0.31$, two-way ANOVA for dose, $n = 4$, F (2, 6) = 1.423), nor fragmentation of sleep-wake as measured by the number of wake bouts over the 6 h after injection ($p = 0.7588$, $n = 4$, $t = 0.3363$, df = 3) nor changes in the duration of longest wake in this period (saline 26 bouts $\pm$ 5.1 versus CNO 0.9 mg/kg 23.75 bouts $\pm$ 2.7, $p = 0.6028$, $n = 4$, $t = 0.5797$, df = 3), which had all been noted after chemogenetic inhibition of all SuM glutamate neurons (Fig. 4). However, SuM$^{Nos1/Vglut2}$ neurons by our estimate account for no more than 20% of the SuM$^{Vglut2}$ population, so inhibition of such a small percentage of SuM$^{Vglut2}$ neurons may not have been sufficient to affect overall wake-sleep. On the other hand, power-spectral analysis revealed a significant decrease in high theta power (7–13 Hz) during REM sleep (Fig. 8i) in these mice when given CNO, but without changes in spectra during wakefulness ($p = 0.1744$, two-way ANOVA, saline versus CNO 0.9 mg/kg, $n = 4$, F (1, 18) = 1.999; Bonferoni post hoc comparisons by band $p > 0.9999$ except for gamma $p = 0.2335$), suggesting a particularly important role for SuM$^{Nos1/Vglut2}$ neurons in normal REM theta activity (Fig. 8l).

## Discussion

Prior studies in humans and animals have indicated that a key, but hitherto uncharacterized, node of the arousal system resides in the caudal hypothalamus[1–16]. Prior neuroanatomical tracing and lesion studies were consistent with the SuM playing a role in arousal, but the results of the present study for the first time demonstrate that the SuM is a major source of ascending arousal tone. Furthermore, we demonstrate that release of glutamate from SuM neurons is necessary for normal amounts of wake and EEG spectral activity in the high-theta and gamma bands and for consolidation of sleep–wake; and that acute inhibition of SuM$^{v-glut2}$ neurons results in an increase in NREM sleep and fragmentation of wakefulness.

With acute inhibition in mice, we also found no changes in REM sleep amounts, although we did note an increase in low frequency REM theta activity when all SuM neurons were inhibited. SuM$^{Nos1/Vglut2}$ neurons were capable of promoting wakefulness when activated, but inhibition of these neurons did not alter sleep-wake or the waking power spectra, although it did result in decreased theta activity in REM sleep. While chronic cell-specific lesions of the SuM in rats have been reported to have little effect on sleep-wake[35, 48] including REM amounts, a recent paper by Renouard et al.[35], reported that SuM lesions reduced 7–10 Hz power in the EEG (high theta in the rat) during REM sleep, and that dentate gyrus cFos expression during REM sleep

**Fig. 5** SuM GABA/glutamate co-releasing neurons. **a** In situ hybridization for vglut2 (magenta) in a vgat-IRES-Cre/L10 reporter mouse (green, see Methods section) reveals vglut2/vgat co-expression (white). **b** These neurons constitute a subpopulation of the larger neurons of the grandicellular SuM (partially opaque). **c** Cartoon showing AAVx-FLEX-ChR2-mCherry recombining in the presence of Cre in either vglut2-Cre or vgat-Cre mice. **d** Cartoon of optogenetically evoked stimulation of SuM terminals on dentate gyrus grandule cells, superimposed on a histological image showing dense anterograde labeling (AAV-FLEX-GFP in vgat-Cre mouse) of SuM terminals in the dentate gyrus supragranular layer and CA2 (dark brown) with a light Nissl counterstain (faint blue–purple; scale bar 500 μm). **e** Representative brain-slice voltage-clamp recording a dentate gyrus granule cell during photostimulation of SuMvgat terminals expressing ChR2-YFP in vgat-IRES-Cre mice (Vh = −70 mV; KCl-based pipette solution; three 5 ms light pulses: blue bars). Photo-evoked postsynaptic currents (average of 30 trials) recorded in control ACSF (Con), in bicuculline-methiodide (Bic, 10 μM), in DNQX (20 nM) and DNQX + Bic. **f** Half-width of GABA-A-mediated and AMPA-mediated photo-evoked postsynaptic currents (30 trials, 3 neurons; mean ± SEM in red; $p < 0.001$ unpaired $t$-test. **g** Photo-evoked (SuMVgat → DG) GABA-A-mediated (left) and AMPA-mediated (right) postsynaptic currents recorded at different membrane potential (Cs-based pipette solutions) and **h** I–V curves of the GABAA- (black circle) and AMPA- (white circle) postsynaptic currents (mean ± SEM, $n = 4$). **i** GABAA- (top trace) and AMPA- (bottom trace) postsynaptic currents evoked by photostimulation of SuMVglut2 → DG input in vglut2-IRES-Cre mice. **j** Onset delay of GABAA- and AMPA-mediated postsynaptic currents (black circle: vgat-IRES-Cre mice and white circle: vglut2-IRES-Cre mice; mean ± SEM in red; $p = 0.002$ paired $t$-test)

was reduced. Based on our data, the loss of SuM[Nos1/Vglut2] neurons may have been responsible for decreased REM theta power.

While the effect of inhibition may appear modest, it was actually striking in relation to the effect of acute inhibition of other components of the arousal system with increased NREM sleep and fragmentation of wakefulness. Inhibition or destruction of other accepted components of the ascending arousal system such as the locus coeruleus, tuberomammillary histamine neurons, midline thalamus, or lateral hypothalamic orexin neurons has little or no effect of sleep wake[18, 21, 35, 48–51] while stimulation in some of these studies can show potent arousal responses[52]. While it is tempting to conclude that the ascending arousal network is highly redundant, it may instead indicate a specificity of function for individual components of the arousal network and it does not mean that these nodes are not playing an important role in intact animals. For example, locus ceruleus acute activation

promotes wakefulness, but destruction of these neurons only prevents wake maintenance in a novel environment, without changing baseline sleep–wake[51].

The role of grandicellular SuM[vgat/vglut2] neurons remains to be determined. They are apparently less involved than the SuM[Nos1] cells in the control of wakefulness, but given the extraordinary density of SuM[vgat/vglut2] terminals on the proximal dendrites of dentate granule cells[42–53] in the supragranular layer, an effect on temporal precision of "dentate gating" and perhaps mnemonic processes appears likely. Furthermore, they may play a role in the plastic changes in the dentate gyrus that occur after status epilepticus and during epileptogenesis, as the SuM terminals sprout and take on an aberrant distribution in the dentate gyrus in a rat pilocarpine model of post-status epilepticus medial temporal lobe epilepsy[53].

Based on the anatomical connectivity of the SuM, as well as these physiological effects, our findings strongly suggest that the

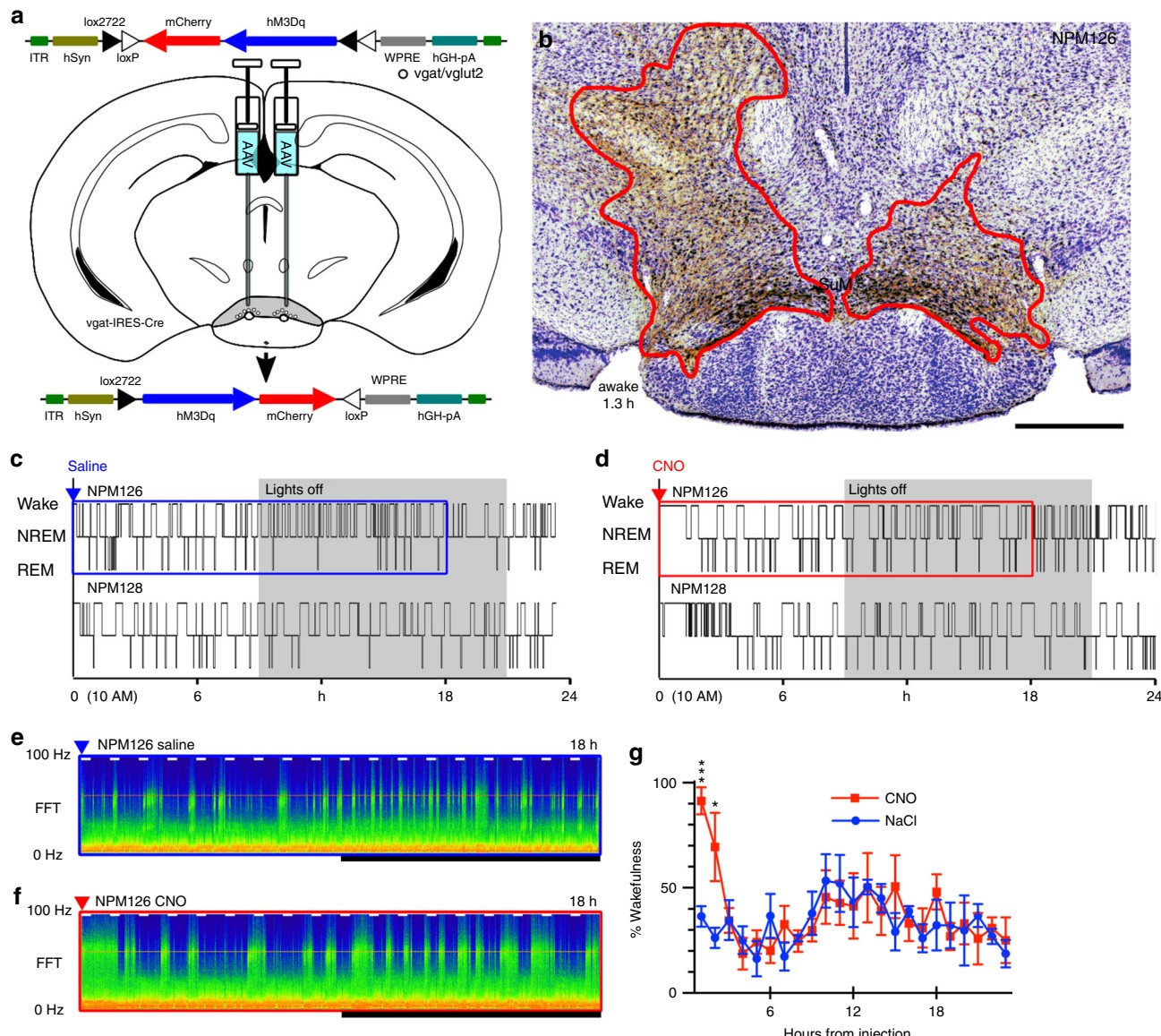

**Fig. 6** Activation of SuM[vgat] neurons results in brief wakefulness. **a** Cartoon showing AAVx-FLEX-hM3Dq-mCherry recombining in vgat neurons that express Cre. **b** An example injection site showing hM3Dq transduction (red outline) in the region containing GABA-glutamate neurons in the SuMg (lower left—duration of wakefulness after CNO in this mouse, scale bar 500 μm). **c**, **d** Hypnograms showing sleep–wake in two mice with saline, then CNO, with a transient increase in wakefulness. **e**, **f** Example CSA's corresponding to the periods shown by the blue and red boxes in **c** and **d**, respectively (0–100 Hz, black bar shows the dark period, short white bars represent 20 min and are spaced 1 h apart)

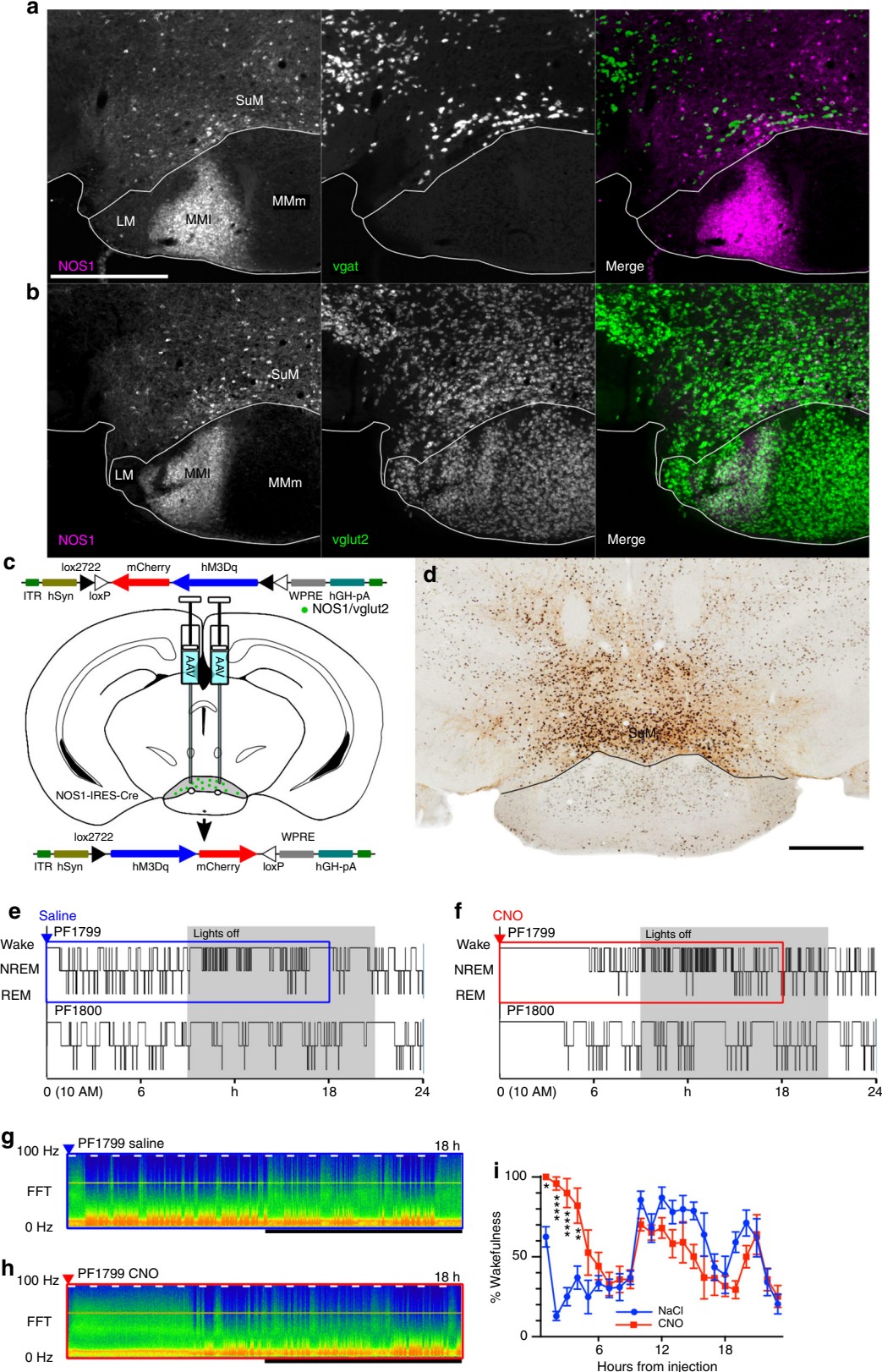

SuM is a key node in the ascending arousal network, but the precise neuroanatomical pathway mediating these wake-promoting effects is not yet proven. Overall, there are three major groups of projections from the SuM: First, the most appreciated output innervates components of the hippocampal network including the recently described entorhinal-dentate-CA2 circuit[54], the septal area and diagonal band, subiculum and entorhinal cortex[20], as well as the medial prefontal cortices[19, 20]. Second, while not as evident in the Vertes[20] study, there is significant topographic innervation of the cerebral cortex[19]. Finally,

particularly in the lateral-most SuM, there are projections to the upper brainstem and the more lateral basal forebrain components of the arousal system[23–25, 55]. Thus, particularly to the lateral SuM, there are strong reciprocal connections with upper brainstem and midbrain components of the arousal system[56], including the parabrachial complex, locus coeruleus, dorsal raphe and laterodorsal tegmental nuclei[20, 23–25, 34, 55, 57–60]. Ascending connections include basal forebrain GABAergic and cholinergic neurons[61], cell groups recently shown to mediate arousal and suppress cortical slowing, respectively[21, 22]. Wakefulness seems most likely due to activation of the second or third group of projections, reliant on the wide distribution of SuM efferents in the cortex and the basal forebrain, similar to those of the locus ceruleus[51]. Notably, prolonged wake was always associated with increased theta activity, so the activation of these pathways by SuM[vglut2] or SuM[Nos1/Vglut2] neurons may not be independent. Consistent with this observation that there are both spatial learning impairments[32, 33] and amnesia[16] after SuM inactivation or lesions in animal models or humans, and, as with the locus ceruleus, reciprocal connection of the SuM with medial prefrontal regions that subserve volitionally directed attention and the volitional maintenance of wakefulness[19, 20, 23, 57]. Our data taken together with previous studies, is most consistent with SuM[vglut2] neurons being the long-sought caudal hypothalamic node of the arousal system, playing a role in wakefulness, as well as attention and mnemonic processes.

## Methods

**Mice.** We used adult male heterozygous *Vglut2-IRES-Cre*, *Vgat-IRES-Cre*, *NOS1-IRES-Cre*, non-*Cre*-expressing littermates and adult male homozygous *Vglut2[lox/lox]* mice (8–12 weeks, 20–25 g; total $n = 58$ in vivo and $n = 8$ in vitro), generated as described elsewhere[62–64] and obtained from the Lowell or Jackson Laboratories. For *Cre* mouse lines, *Cre*-recombinase was targeted just distal to the stop codon of the *vgat* or *vglut2* genes, respectively, using an optimized internal ribosomal entry sequence (IRES), with endogenous gene promoters therefore driving Cre-recombinase expression via a bicistronic message[62]. *NOS1-IRES-Cre* mice were generated by insertion of an internal ribosome entry site (IRES) along with *Cre* recombinase into the 3′ untranslated region of *Nos1*[63]. *Vglut2* conditional knockout mice have *loxP* sites flanking the second exon of the *Vglut2* gene (*Vglut2[flox/flox]*)[64]. Each strain was originally derived from *129S* mice or embryonic stem cells, but were back-crossed with *C57BL/6J* mice for two or more generations. All mice were bred and genotyped, before and after use, in-house and maintained on a 12 h light–dark cycle with food and water ad libitum. Procedures met National Institutes of Health and Guide for the Care and Use of Laboratory Animals standards; all protocols were approved by the Beth Israel Deaconess Medical Center IACUC.

**Vectors.** We used the *pAAV-hSyn-DIO-hM3D-mCherry* and *pAAV-hSyn-DIO-hM4D-mCherry* plasmids (from Dr. Brian Roth, University of North Carolina) to provide the *Cre*-dependent hM3Dq and hM4Di transgene, packaged by standard triple transfection protocol to generate helper virus-free pseudotyped AAV2/10 virus. An AAV2/10 rep/cap plasmid provided AAV2 replicase and AAV10 capsid functions, while adenoviral helper functions were supplied by the pHelper plasmid (Stratagene, La Jolla, CA). Briefly, AAV-293 cells were transfected via calcium phosphate precipitation with 1.33 pmol of pHelper, and 1.15 pmol each of AAV2/10 and an AAV vector plasmid (double-floxed or FLEXED configuration[45]). The cells were collected 72 h later and the pellets resuspended in DMEM, freeze-thawed three times and centrifuged to produce a clarified viral lysate. Vector stocks were titered by rtPCR (Eppendorf Realplex), ranging from ~ $1 \times 10^{12}$ to $1 \times 10^{13}$ copies/ml. The necessity for Cre-enablement for hM3Dq and hM4Di expression was confirmed in vitro using HEK293-Cre cells. These vectors are referred to as AAV-hM3Dq and AAV-hM4Di. For experiments in vglut2flox/flox mice, we used an AAV serotype 10 vector expressing Cre-recombinase fused to GFP (AAV-Cre; Dr Caroline Bass, University at Buffalo).

**Surgery.** Mice were anesthetized with ketamine/xylazine (100/10 mg/kg i.p.) and placed in a stereotaxic frame. To selectively transduce hM3Dq receptors in glutamatergic or GABAergic neurons of the SuM (SuM[vglut2] and SuM[vgat], respectively), we placed bilateral injections of AAV-hM3Dq or AAV-hM4Di (15–75 nl/side into the caudal hypothalamus (coordinates: AP = −2.45 to −2.7 mm, $L = \pm 0.4$ to 0.5 mm, DV = −4.9 mm[65]) of Vgat-IRES-Cre and Vglut2-IRES-Cre mice. For controls we injected AAV-hM3Dq and AAV-hM4Di in non-Cre-expressing littermates, with no changes in sleep–wake physiology, nor transduction of hM3Dq or hM4Di, as we have previously reported[21, 66]. To transduce hM3Dq only in neurons with disrupted vglut2 expression, we mixed equal portions of AAV-Cre and AAV-hM3Dq, injecting this as above into the SuM of *vglut2[lox/lox]* mice. Microinjections were made using a compressed air delivery system[67]. After injections, headsets were implanted, consisting of a six pin plug (Heilind Electronics, Inc., Willmington, MA) soldered to four EEG screws (Pinnacle Technologies, Lawrence, KS) and two silver pad electrodes, then secured with dental cement. Electrodes were 1 mm rostral and lateral to Bregma, 2 mm caudal and lateral to Bregma, with EMG electrodes over nuchal muscles. In some, a twisted-pair electrode was placed in the dorsal hippocampus unilaterally, with contacts either side of the dentate gyrus pyramidal cell layer (Plastics One, Wallingford, CT). Mice were kept warm until normal activity resumed then housed singly thereafter.

**Sleep–wake recordings.** Four to 6 weeks later, mice were placed in transparent acrylic barrels in well-ventilated, light-tight and sound attenuating isolation chambers on a 12 h light–dark cycle and at 22 ± 1 °C, with typical bedding material and ad libitum food and water. Mice were connected to a commutator with a preamplifier unit (8202SL, Pinnacle Technologies, Lawrence, KS), and habituated for 72 h. Amplifier units (8200-K1-SL) were connected to a recording computer running Sirenia Acquisition (Pinnacle Technologies) acquiring EEG (one oblique (frontal contralateral parietal) and one longitudinal (frontal parietal) derivation, and bipolar nuchal EMG), sampled at 400 or 600 Hz, with a hardware high-pass filter of 0.5 Hz, most with synchronized infrared video. CNO and vehicle were given in random order in early experiments with large sites of vector transduction (Fig. 1c–f), and no order effect was noted.

**Histology.** Prior to perfusion, mice received CNO (0.3 mg/kg, IP 10 AM) before deep anesthesia with chloral hydrate (200 mg/kg) and transcardial perfusion (20 ml saline then 100 ml of neutral phosphate-buffered formalin (4%, Fisher Scientific, Pittsburgh, PA). Brains were removed, incubated in 20% sucrose at 4 °C for at least 48 h, then sectioned at 40 μm on a freezing microtome in three series. cFos and DsRed immunohistochemical staining used a diaminobenzidine (DAB) reaction, with sections incubated overnight with primary (1:20 K, cFos; 1:10 K DsRed) then secondary antibodies for 2 h, ABC reagents (1:1000; Vector Laboratories) for 90 min, then a 0.06% solution of 3,3-diaminobenzidine tetrahydrochloride (Sigma-Aldrich), sometimes with 0.05% $CoCl_2$ and 0.01% $NiSO_4$ ($NH_4$), in PBS plus 0.02% $H_2O_2$ for about 5 min (based on observed reaction product). The rabbit polyclonal Fos antibody was raised against synthetic human cFos (4–17) (Oncogene Sciences, rabbit polyclonal cat#Ab5). Rabbit polyclonal antibody anti-mCherry was raised against DsRed (Clontech cat#632496) with specificity indicated by the lack of immunostaining in control mice. For secondary antibody immunohistochemical controls, primary antibodies were omitted and the tissue showed no background immunoreactivity. Secondary antibodies included donkey anti-rabbit alexa fluor 546 (Invitrogen; 1:500) and donkey anti-rabbit biotinylated (Invitrogen; 1:500).

In situ hybridization revealed the distribution of SuM neurons co-expressing markers of GABA and glutamate release[42, 43], using a vgat-IRES-Cre mouse, crossed with an L10 reporter line[68], along with in situ hybridization for vglut2 RNA. Sections were incubated overnight with hybridization buffer containing vglut2 probe at 52.5 °C (gift from Dr. Shigafumi Yokota, Shimane University, Izumo, Japan; see Niu et al.,[60] for sense controls), rinsed in standard saline citrate (SSC) with 50% formamide after rinsing in Tris-buffered saline (TBS) pH 7.5.

**Fig. 7** Activation of SuM[Nos1] neurons promotes wakefulness. **a** NOS1 immunohistochemisty (left, white scale bar 500 μm applies to all panels of **a** and **b**) reveals a subpopulation of SuM neurons that only rarely contain vgat, shown as gray-white neurons on the merged image (right). **b** NOS1 immunohistochemistry (left) in a vglut-Cre L10 reported mouse (center) showing that NOS1 neurons are a subset of vglut2 neurons, with no examples of neurons with NOS immoreactivity that were not vglut2 positive (right, merge). **c** Cartoon showing AAVx-FLEX-hM3Dq-mCherry recombining in NOS1 Cre expressing neurons. **d** Injection site showing DAB reaction product in neurons positive for mCherry, with black nuclear reaction product in cFos positive neurons after CNO injection (black scale bar, bottom right, 500 μm). **e**, **f** Hypnograms from two mice after vehicle and CNO, shows a strong effect of NOS1 cellular activation on wakefulness, with **g**, **h** the CSA (0–100 Hz, black bar shows the dark period, short white bars represent 20 min and are spaced 1 h apart) from the areas shown in **e** and **f** by the blue and red boxes, respectively. **i** Pooled data from seven mice showing the percentage of wakefulness in 1 h bins after injection (****$p < 0.0001$, **$p = 0.0023$, *$p = 0.0248$)

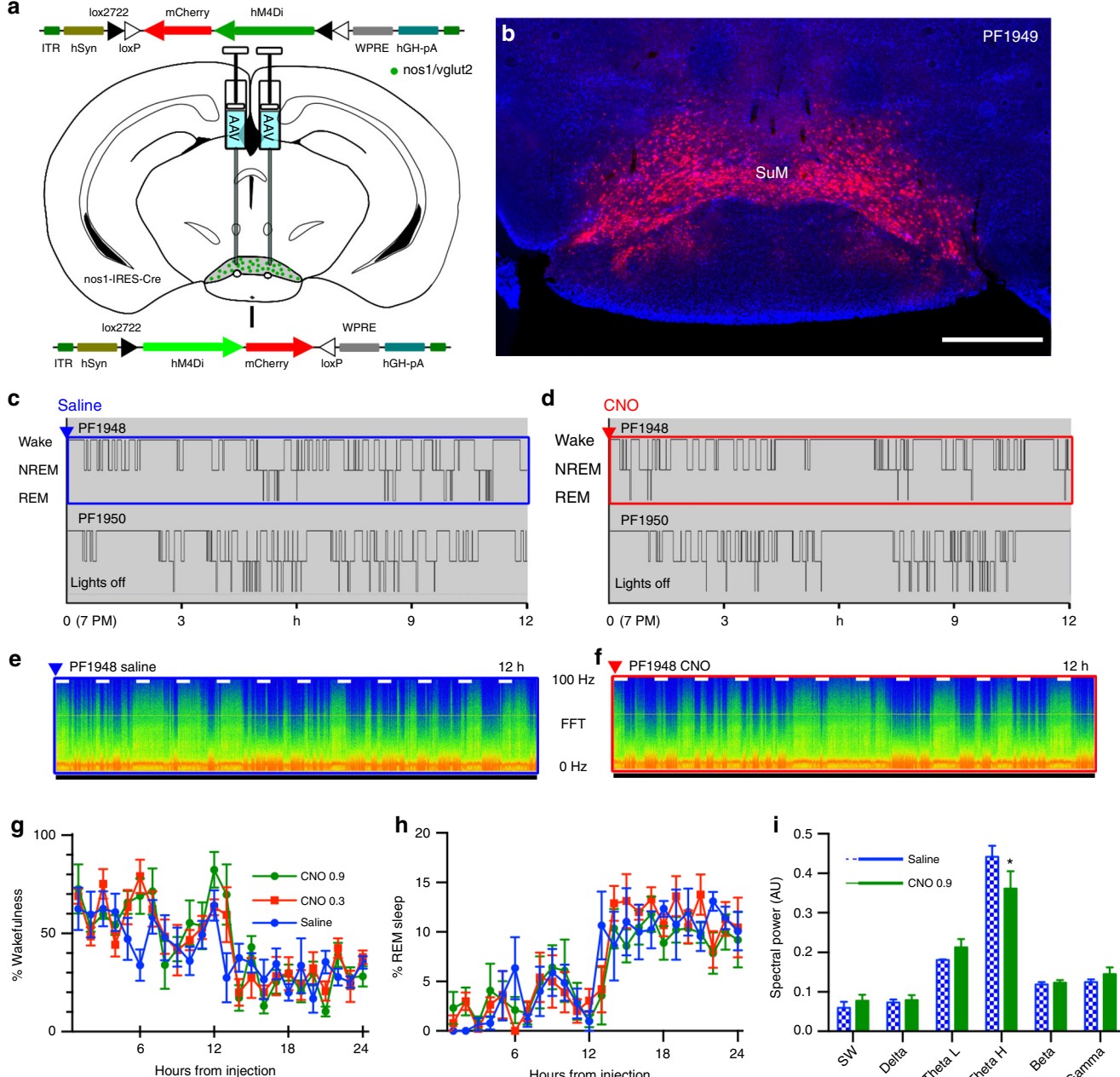

**Fig. 8** Inhibition of SuM$^{Nos1}$ does not cause somnolence, but reduces REM theta activity. **a** Cartoon showing AAV-FLEX-hM4Di-mCherry recombining in Nos1 neurons that express Cre. **b** Conditional transduction of inhibitory hM4Di in Nos1-IRES-Cre mice (mCherry fluorescence with DAPI counterstain; scale bar 500 μm). **c**, **d** Hypnograms from two mice over 12 h after vehicle and CNO 0.9 mg/kg injected at ZT12 (7 PM, lights-off). Unlike inhibition of all glutamate neurons (Fig. 4), the hypnograms appeared similar regardless of treatment. **e**, **f** CSA after vehicle and CNO from the periods shown by the blue and red boxes, respectively, in **c** and **d** (0–100 Hz, black bar shows the dark period, short white bars represent 20 min and are spaced 1 h apart). **g**, **h** There is no change in the amount of wakefulness, NREM or REM sleep (separate two-way ANOVA, wake, REM, and NREM: F (2, 192) = 1.709, 0.8381, 2.04; p = 0.1837, 0.4341, 0.1329, respectively, n = 5). **i** Power spectra (band mean ± SEM) were only changed for REM sleep in post hoc comparisons, with a significant suppression of waking high theta activity (Bonferroni *p = 0.0332, all other bands p > 0.9999, n = 3, t = 3.048, df = 24)

After additional TBS rinses, sections were incubated for 30 min in 1% blocking reagent (Roche, cat#11 096 176 001) then overnight in peroxidise conjugated anti-digoxigenin antibody (1:500; Roche cat#11 207 733 910). Sections were rinsed in TBS then reacted with 1:50 Tyramide signal amplification fluorsent kit (TSA Plus-A-Cy3, Perkin Elmer cat#NEL744001KT) diluted in amplification diluent (30 min).

To determine neuronal size, one high power field of the SuM, immediately above the mammillothalamic tract before it emerges dorsally through the mammillary capsule was use to compare the size of SuM$^{vgat/vglut2}$ and SuM$^{vglut2}$ neurons. Neurons with clear nuclei and unambiguously distinguishable from adjacent neurons were included and measured through their long axis. Student's t-test was used for parametric two-tailed comparison.

**Data analysis**. Physiological Data Analysis was performed with Spike 2 (version 8.05, Cambridge Electronic Design, UK) using the Ratsleepauto script (version 2.02, Cambridge Electronic Design), with complete manual rescoring to correct any errors. Compressed spectral arrays were created by use of the sonogram feature of Spike 2 (settings 32768 points, Hanning window). Images of sonograms were imported into FIJI (ImageJ 2.0.0, NIH) despeckled before transfer to Inkscape (version 0.91) for figure creation. Spectral data for comparison of discrete spectral bands was performed by 'Ratsleepauto' script with 1024 points per Hanning window, saved as text and imported to Excel where 60 Hz artifact was removed by interpolation of points from 59.5 to 60.5 Hz, and spectral values were added into spectral bands (slow wave 0–2 Hz, delta 2–4 Hz, low theta (LTheta) 4–7 Hz, high theta (HTheta) 7–13 Hz, beta 13–30 Hz, gamma 30–120 Hz). Statistical analysis

used two-way repeated-measures ANOVA for analysis of spectral and sleep–wake effects. Post hoc and multiple $t$-tests were corrected with Bonferroni's method. Student's $t$-test was used when a single comparison was made, and was two-tailed in all cases. Software used was Prism (5.0, 6.0 h, 6.0 g; GraphPad, La Jolla, CA).

For objective statistical analysis of injection sites, a series of 16.4 T MRI gradient echo images of the C57 mouse brain were obtained[69], resampled to 30 (lateral) by 30 (dorsoventral) by 120 (axial) micron voxels and rotated to match the our typical plane of histological sections and sampling (Mango version 3.4, Research Imaging Institute, University of Texas). Injection sites were plotted (MRICron[70] (version 8/4/2014), McCausland Center for Brain Imaging, University of South Carolina)— when staining could not be decided between somatic and parenchymal were included in the injection site. Binary image, continuous group voxel-based lesion-symptom mapping analysis with 1000 permutations was then conducted (NPM (version 6/1/2015) component of MRICron) to correlate the duration of wakefulness with injection sites, with negative $z$-scores then used (given association with increasing score) corresponding to a false discovery rate $\leq 1\%$, and plotted as a color map on MRI sections, as is standardly used for lesion-symptom mapping in human fMRI studies[71].

**Image processing and graphics**. Light micrographs (Zeiss Axioplan 2 or Olympus Slide Scanner) were adjusted with respect to color balance, background and contrast (FIJI, ImageJ 2.0.0 with BIOP VSI Reader Pluggin, NIH), rotated and cropped (Gimp 2.8.14) and imported into Inkscape (version 0.91). Fluorescence images (Zeiss Axioplan 2 or Olympus Slide Scanner) were recolored as magenta (vglut2 in situ) and green (vgat-L10-GFP reporter). Brightness and contrast were adjusted for each channel to make regions outside of the brain tissue black, and apply maximal brightness to the highest signal. For Fig. 5a images were combined and a threshold function was applied to recolor as white regions that had high signal in both green and magenta channels to enhance the visibility of clearly double-labeled neurons, with the original combined images serving as a reference (FIJI, ImageJ 2.0.0, NIH). Images were rotated and cropped (Gimp 2.8.14), before being imported into Inkscape (Inkscape v0.91).

**In vitro experiments**. vgat-IRES-Cre ($n = 5$) and vglut2-IRES-Cre ($n = 8$) mice (8–12 weeks, 20–25 g) were used for in vitro electrophysiological recordings. Mice were injected unilaterally in the SuM (bregma −2.55 mm; midline 0.4 mm; dorsal surface 4.9 mm) with 75 nl of an adeno-associated viral (AAV) vector coding for Cre-dependent channelrhodopsin-2 (ChR2) (AAV-Flex-ChR2(H134R)-YFP vector; titer: $6 \times 10^{12}$ genomic copies/ml; circuit mapping experiments) or with 75 nl of AAV-hM4Di-mCherry or AAV-Flex-mCherry (in vitro CNO experiments). Four to six weeks after AAV injections, we performed in vitro recordings with further verification of the injection site after recording by fluorescent microscopy.

To prepare brain slices, mice were anesthetized with isoflurane (5% in $O_2$) and transcardially perfused with ice-cold ACSF (N-methyl-D-glucamine, NMDG-based solution) containing (in mM): 100 NMDG-Cl, 2.5 KCl, 20 HEPES, 1.24 NaH$_2$PO$_4$, 30 NaHCO$_3$, 25 glucose, 2 thiourea, 5 Na-ascorbate, 3 Na-pyruvate, 0.5 CaCl$_2$, 10 MgSO$_4$ (pH 7.3 when carbogenated with 95% $O_2$ and 5% $CO_2$). Brains were quickly removed and cut in coronal slices (250 μm thick) using a vibrating microtome (VT1200S, Leica, Bannockburn, IL, USA), then transferred into normal ACSF (Na-based solution) containing (in mM): 120 NaCl, 2.5 KCl, 1.3 MgSO$_4$, 10 glucose, 26 NaHCO$_3$, 1.24 NaH$_2$PO$_4$, 2 thiourea, 1 Na-ascorbate, 3 Na-pyruvate, 4 CaCl$_2$, (pH 7.4 when carbogenated with 95% $O_2$ and 5% $CO_2$, 310–320 mOsm). For circuit mapping experiments, we recorded from granule cells in the dentate gyrus of the dorsal hippocampus (bregma: −1.7 to −2.3 mm)[65], targeted by use of infrared differential interference contrast (IR-DIC) video microscopy and IR-sensitive CCD camera (ORCA-ER, Hamamatsu, Bridgewater, NJ, USA). Whole-cell recordings used a Multiclamp 700B amplifier (Molecular Devices, Foster City, CA, USA), a Digidata 1550 A interface and Clampex 10.6 software (Molecular Devices). SuM axons and synaptic terminals expressing ChR2 were activated by a full-field 5 ms flashes of light ($\sim$ 10 mW/mm$^2$ 1 mm beam width) from a 5 W luxeon blue light-emitted diode (470 nm wavelength; #M470L2-C4; Thorlabs, Newton, NJ, USA) coupled to the epifluorescence pathway of the Zeiss microscope. The area stimulated included the recorded cell, centered in a 500 μm radius concentric field. Photo-evoked postsynaptic currents were recorded using a KCl-based pipette solution containing (in mM): 140 KCl, 1 EGTA, 10 HEPES, 1 MgCl$_2$, 5 Mg-ATP, and 0.3 Na-GTP (pH adjusted to 7.2 with KOH, 280 mOsm) or a Cs-based pipette solution containing (in mM): 125 CsMeS, 11 KCl, 10 HEPES, 0.1 CaCl$_2$, 1 EGTA, 5 Mg-ATP, and 0.3 Na-GTP (pH adjusted to 7.2 with KOH, 280 mOsm). Electrophysiological data were analyzed using Clampfit 10.5 (Molecular Devices) and IGOR Pro 6 (WaveMetrics, Lake Oswego, OR, USA). Synaptic events were detected off-line using Mini Analysis 6 (Synaptosoft, Leonia, NJ, USA). The latency of the photo-evoked postsynaptic currents was determined by the time difference between the starting of the light pulse and the 5% rise point of the first synaptic event[72]. Results are expressed as mean ± SEM, $n$ refers to the number of cells. For the experiments with CNO we recorded SuM$^{vglut2}$ neurons expressing hM4Di-mcherry (transfected with the AAV-hM4Di-mCherry vector) or only mCherry (transfected with the AAV-Flex-mCherry vector; control). Current-clamp and cell attached recordings were made using K-Glu-based pipette solution containing (in mM): 120 κ-gluconate, 10 KCl, 3 MgCl$_2$, 10 HEPES, 2.5 K-ATP, and

0.5 Na-GTP (pH adjusted to 7.2 with KOH, 280 mOsm). CNO was dissolved in normal ACSF to final concentration of 5 μM on the day of the experiment. All drugs used in the in vitro experiments were purchased from Thermo Fisher Scientific (Waltham, MA) with the exception of Bicuculline methiodide (Tocris, Bristol, UK) and CNO (Sigma-Aldrich, St. Louis, MO).

**Data availability**. All data are available by contacting the corresponding author at npeders@emory.edu.

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

## Acknowledgements

We are grateful for the technical assistance of Quan Hue Ha, Minh Ha, Sarah Keating, Myriam Debryune, and Rebecca Broadhurst, and for the kind gift, from Drs Bradford Lowell and Linh Vong, of *Vgat-IRES-cre* and *Vglut2-IRES-cre* lines of mice. This work was supported by the National Institutes of Health R25NS070682 and an American Academy of Neurology and American Brain Foundation Clinician-Researcher Training Fellowship (N.P.P.), as well as NARSAD Young Investigator Grant (A.V.), National Health and Medical Research Council of Australia Early Career Fellowship GNT1052674 (S.B.G.A.). National Health and Medical Research Council C.J. Martin Award (S.B.G.A.), National Institutes of Health grants R21NS082854 (E.A.), R01NS073613 (P.M.F.), R01NS092652 (P.M.F.), R01NS085477 (C.B.S.), and P01HL095491 (C.B.S., E.A.).

## Author contributions

N.P.P., N.V., E.A., P.M.F. and C.B.S. were responsible for study inception and design of experiments. N.P.P., L.F., J.L.W., A.V., S.B.A., E.A. and P.M.F. performed experiments.

N.P.P., L.F., J.L.W., A.V., S.B.A., E.A., P.M.F. and C.B.S. analyzed or interpreted data. N.P.P., E.A., P.M.F. and C.B.S. wrote the paper.

## Additional information

**Competing interests:** The authors declare no competing financial interests.

