## [Peer Review File · Nature Communications]

Reviewers' comments:

Reviewer #1 (Remarks to the Author):

This is the second time I have reviewed this manuscript. The authors have now satisfactorily addressed my comments in their revised and improved manuscript. I think this is an in a interesting and original study whose results will interest a wide range of researchers. The identification of glutamate-projection neurons from the caudal hypothalamus, some of which also use NO is an important discovery. This clearly extends our picture of arousal-promoting neurones beyond histamine and orexin neurons from this region of the brain. The future challenge is to work out how these pathways interact. The identification of glutamate-GABA co-releasing neurons is also of interest, even if it is less clear what this subgroup of neurons is contributing to function.

There is enough detail in the manuscript for other researchers to reproduce the work.

William Wisden

Reviewer #4 (Remarks to the Author):

In the present work, the authors revealed that the glutamate-releasing neurons of the supramammillary region (SuM) are critical in producing behavioral and EEG arousal, and should work as a key node for the wake-sleep regulatory system. Using a combination of targeted genetic and viral vector technologies in mice, they were able to show three subpopulations of glutamate cells in the SuM, and provided clear data characterizing how each one of the SuM glutamatergic cell groups contributes to cortical and hippocampal activation and behavioral state. The results are exceedingly important, and pointed out for the first time the SuM as a key element critical for the arousal system. The work was masterfully performed, and I have just a couple questions, which I would like to be addressed.

The authors described that following selective genetic disruption of glutamate release from the SuM there was little effect on the amounts of wake, REM and NREM sleep in behaving animals. Considering the importance of this hub in the maintenance of arousal, as suggested by the authors, how on the long run the lack of this glutamatergic node could be compensated and result in a normal wake-sleep cycle.

Notably, as can be depicted from the work of Vertes (J. Comp. Neurol. 1992; 326:595-622), the main targets of the SuM are sites related to the hippocampal formation (i.e., the septal area, the hippocampus proper and nucleus reuniens). In contrast, weaker projections were noted to the basal forebrain and only very sparse inputs to cortical areas. Considering this scenario, the authors should discuss how the SuM, here postulated as a critical node to control arousal, seems to have a larger impact on the hippocampal formation than in the rest of the cortex.

Reviewer #1 (Remarks to the Author):

We are grateful for the time and comments of Reviewer #1, Dr. Wisden, that enabled the prior improvement of our submission. There are no required changes based on this review of our updated manuscript.

“This is the second time I have reviewed this manuscript. The authors have now satisfactorily addressed my comments in their revised and improved manuscript. I think this is an in a interesting and original study whose results will interest a wide range of researchers. The identification of glutamate-projection neurons from the caudal hypothalamus, some of which also use NO is an important discovery. This clearly extends our picture of arousal-promoting neurones beyond histamine and orexin neurons from this region of the brain. The future challenge is to work out how these pathways interact. The identification of glutamate-GABA co-releasing neurons is also of interest, even if it is less clear what this subgroup of neurons is contributing to function.

“There is enough detail in the manuscript for other researchers to reproduce the work.”

Reviewer #4 (Remarks to the Author):

We are thankful for the time and comments of Reviewer #4. There are two key and important questions raised, appearing as paragraphs three and four of the review below. Please see our in-line responses and tracked changes in our revised manuscript.

“In the present work, the authors revealed that the glutamate-releasing neurons of the supramammillary region (SuM) are critical in producing behavioral and EEG arousal, and should work as a key node for the wake-sleep regulatory system. Using a combination of targeted genetic and viral vector technologies in mice, they were able to show three subpopulations of glutamate cells in the SuM, and provided clear data characterizing how each one of the SuM glutamatergic cell groups contributes to cortical and hippocampal activation and behavioral state. The results are exceedingly important, and pointed out for the first time the SuM as a key element critical for the arousal system.

“The work was masterfully performed, and I have just a couple questions, which I would like to be addressed.

“The authors described that following selective genetic disruption of glutamate release from the SuM there was little effect on the amounts of wake, REM and NREM sleep in behaving animals. Considering the importance of this hub in the maintenance of arousal, as suggested by the authors, how on the long run the lack of this glutamatergic node could be compensated and result in a normal wake-sleep cycle.

We agree with the reviewer, and have rewritten the relevant part of the discussion to reflect this. A number of key arousal-promoting cell groups (e.g., the locus coeruleus, orexin neurons, basal forebrain cholinergic neurons) that cause arousal when stimulated acutely, have little effect on baseline wake-sleep amounts when chronically lesioned. This may reflect the ability of the nervous system to reorganize and compensate after acute lesions, but may also reflect the lack of sophistication in what we measure. For example, after LC lesions, animals cannot maintain wakefulness in response to novel stimuli (Gompf et al., 2010), and animals without orexin neurons have narcolepsy. So,

further testing may be necessary to identify a phenotype, beyond the described sleep-wake effect, for the animals with acute inhibition of the SUM glutamatergic neurons. On the other hand, lesions of the posterior lateral hypothalamus in both animals and humans do produce a state of prolonged sleepiness. How much of this is caused by damage to the SUM neurons, or to the tuberomammillary histamine, or lateral hypothalamic orexin or GABA neurons now known to promote wake (Venner et al, 2017; Herrera et al., 2016) in the area, or to fibers of passage from the parabrachial nucleus or locus coeruleus or dopamine neurons in the brainstem again will require further research. However, our paper adds another important piece to the puzzle.

“Notably, as can be depicted from the work of Vertes (J. Comp. Neurol. 1992; 326:595-622), the main targets of the SuM are sites related to the hippocampal formation (i.e., the septal area, the hippocampus proper and nucleus reuniens). In contrast, weaker projections were noted to the basal forebrain and only very sparse inputs to cortical areas. Considering this scenario, the authors should discuss how the SuM, here postulated as a critical node to control arousal, seems to have a larger impact on the hippocampal formation than in the rest of the cortex.”

We agree that we do not yet know which projection from the SUM accounts for the arousal response, and now point out in the text that this will require more investigation. However, we would not write off a direct cortical projection based on the Vertes (1992) paper. In Saper (1985), retrograde transport studies showed that SUM neurons project to the entire cortical surface in a topographic manner. These projections to the lateral wall of the hemisphere come from the far lateral SUM (which Saper called the posterior lateral hypothalamus, although clearly meant the same area). This part of the SUM was not really included in the Vertes paper (the most lateral injection was quite far caudal (right panel below), involving mainly the grandicellular Vglut2/Vgat neurons (left panel) that project largely to the hippocampal formation). The far lateral SuM region innervates the parabrachial region (Swanson, 1982) and substantia innominata (Grove, 1988), providing a substrate for activation of the upper brainstem and basal forebrain to promote wakefulness. Using retrograde tracing from the basal forebrain, Vertes (1988) has also shown innervation of the substantia innominata and magnocellular preoptic area by the lateral SuM, with retrograde labeling from the septal area and diagonal band arising from the more medial SuM.

[Redacted]

Top left: Our Figure showing vgat, vglut2 neurons and overlap (white) in the grandicellular area. Top center: From Vertes (1992) showing the lateral SuM injection site, not including the far lateral region and situated in the very caudal SuM. Top right: The location of terminals after anterograde labeling from the parabrachial region (Swanson, 1982). Bottom left: retrograde labeling in the SuM after retrograde tracer injection into the magnocellular preoptic and substantia innominata (Vertes, 1988). Bottom right: Retrogradely labeled neurons in the lateral SuM after tracer injection in the substantia innominata (Grove, 1988).

Our own work is consistent with these observations. We have anterogradely labeled the SUM projections to with AAV-GFP in Vglut2-Cre mice, and find that there is actually a rather strong projection to the basal forebrain (below panel E), and considerably more widespread and dense cortical projection (D) from the SUM than the Vertes (1992) paper suggested. Again, we now discuss this, but prefer to reserve judgment on the SUM targets most important for arousal pending future investigation.

[Redacted]

1. Gompf, H. S. *et al.* Locus ceruleus and anterior cingulate cortex sustain wakefulness in a novel environment. *Journal of Neuroscience* **30**, 14543–14551 (2010).
2. venner, A., Anaclet, C., Broadhurst, R. Y., Saper, C. B. & Fuller, P. M. A Novel Population of Wake-Promoting GABAergic Neurons in the Ventral Lateral Hypothalamus. *Curr. Biol.* (2016). doi:10.1016/j.cub.2016.05.078
3. Herrera, C. G. *et al.* Hypothalamic feedforward inhibition of thalamocortical network controls arousal and consciousness. *Nat Neurosci* **19**, 290–298 (2016).
4. Vertes, R. P. PHA-L analysis of projections from the supramammillary nucleus in the rat. *J. Comp. Neurol.* **326**, 595–622 (1992).
5. saper, C. B. Organization of cerebral cortical afferent systems in the rat. II. Hypothalamocortical projections. *J. Comp. Neurol.* **237**, 21–46 (1985).
6. Swanson, L. W. The projections of the ventral tegmental area and adjacent regions: a combined fluorescent retrograde tracer and immunofluorescence study in the rat. *Brain Res Bull* **9**, 321–353 (1982).
7. Grove, E. A. Neural associations of the substantia innominata in the rat: afferent connections. *J. Comp. Neurol.* **277**, 315–346 (1988).
8. Vertes, R. P. Brainstem afferents to the basal forebrain in the rat. *NSC* **24**, 907–935 (1988).

REVIEWERS' COMMENTS:

Reviewer #4 (Remarks to the Author):

The authors addressed very nicely all the issues raised by this reviewer.